# Zooming in on Long Non-Coding RNAs in Ewing Sarcoma Pathogenesis

**DOI:** 10.3390/cells11081267

**Published:** 2022-04-08

**Authors:** Dave N. T. Aryee, Valerie Fock, Utkarsh Kapoor, Branka Radic-Sarikas, Heinrich Kovar

**Affiliations:** 1St. Anna Children’s Cancer Research Institute, 1090 Vienna, Austria; valerie.fock@ccri.at (V.F.); utkarsh.kapoor@ccri.at (U.K.); branka.radic-sarikas@ccri.at (B.R.-S.); heinrich.kovar@ccri.at (H.K.); 2Department of Pediatrics, Medical University of Vienna, 1090 Vienna, Austria; 3Department of Pediatric Surgery, Medical University of Vienna, 1090 Vienna, Austria

**Keywords:** Ewing sarcoma, long non-coding RNAs, competing endogenous (ce) RNA, regulatory RNA, biomarkers, therapeutic targets

## Abstract

Ewing sarcoma (ES) is a rare aggressive cancer of bone and soft tissue that is mainly characterized by a reciprocal chromosomal translocation. As a result, about 90% of cases express the EWS-FLI1 fusion protein that has been shown to function as an aberrant transcription factor driving sarcomagenesis. ES is the second most common malignant bone tumor in children and young adults. Current treatment modalities include dose-intensified chemo- and radiotherapy, as well as surgery. Despite these strategies, patients who present with metastasis or relapse still have dismal prognosis, warranting a better understanding of treatment resistant-disease biology in order to generate better prognostic and therapeutic tools. Since the genomes of ES tumors are relatively quiet and stable, exploring the contributions of epigenetic mechanisms in the initiation and progression of the disease becomes inevitable. The search for novel biomarkers and potential therapeutic targets of cancer metastasis and chemotherapeutic drug resistance is increasingly focusing on long non-coding RNAs (lncRNAs). Recent advances in genome analysis by high throughput sequencing have immensely expanded and advanced our knowledge of lncRNAs. They are non-protein coding RNA species with multiple biological functions that have been shown to be dysregulated in many diseases and are emerging as crucial players in cancer development. Understanding the various roles of lncRNAs in tumorigenesis and metastasis would determine eclectic avenues to establish therapeutic and diagnostic targets. In ES, some lncRNAs have been implicated in cell proliferation, migration and invasion, features that make them suitable as relevant biomarkers and therapeutic targets. In this review, we comprehensively discuss known lncRNAs implicated in ES that could serve as potential biomarkers and therapeutic targets of the disease. Though some current reviews have discussed non-coding RNAs in ES, to our knowledge, this is the first review focusing exclusively on ES-associated lncRNAs.

## 1. Introduction

Ewing sarcoma (ES) is one of the three most common primary bone cancers in addition to osteosarcoma and chondrosarcoma. While chondrosarcoma is supposed to originate from growthplate chondrocytes and osteosarcoma from pre-osteoblasts or osteoblasts, the histogenetic and developmental origins of ES are still unknown. In contrast to chondrosarcoma, which affects mostly middle-aged and older adults, and similar to osteosarcoma, the peak incidence for ES is in adolescence. In spite of the fact that ES presents a quiet genome with significant epigenetic heterogeneity different from osteosarcoma and chondrosarcoma, ES is unique on the transcriptomic level with some similarity but still distinct from mesenchymal stem cells. ES, the second most common pediatric bone cancer, is characterized by a chromosomal translocation that mostly results in the fusion between the EWS breakpoint region 1/EWS RNA binding protein 1 (EWSR1) gene on chromosome 22 and an E26 transformation-specific (ETS) family transcription factor gene, either FLI1 at chromosome 11q24 or ERG at 21q11 [1,2]. In rare cases (<1%), EWSR1 is fused to alternative ETS genes FEV, ETV1, or ETV4 [3]. The EWSR1::ETS oncogenic fusion is so far the only recurrent and tumor specific genetic alteration in ES. It encodes an aberrant transcription factor harboring an ETS DNA-binding domain fused to the low complexity N-terminal domain of the RNA-binding protein EWS. The functional contribution of EWS to EWS-FLI1 mechanism of action is only partially characterized and has so far been associated predominantly with transcriptional activation. The chimeric EWS-ETS protein plays a crucial role in the pathogenesis and proliferation of the disease and its knockdown results in decreased cell proliferation in vitro and tumor regression in vivo [4]. Treatment of ES in North America and Europe and the roles of various therapies as well as relevant guidelines have recently been elaborately reported in a review [5]. Classical treatments of ES have included aggressive neo-adjuvant and adjuvant chemotherapy in combination with surgery and/or radiotherapy, which have enhanced the long-term survival of patients suffering from localized disease, with a 5-year survival of more than 70% [4]. However, when the tumor cells have metastasized or recurred, outcomes for patients are generally poor warranting the search for more efficient treatment strategies. Systemic tumor control still poses the major therapeutic hurdle in ES [4].

Advances in RNA-seq data analyses reveal that 80–85% of the human genome is actively transcribed and a major part of the transcripts are non-coding RNAs (ncRNAs) [6]. Though they are generally not translated into proteins, ncRNAs have been shown to impact gene expression by regulating transcription, post-transcriptional modifications as well as translation [7,8], and their alteration has been implicated in the pathogenesis of several diseases, including cancer. Long non-coding RNAs (lncRNAs), defined as ncRNA transcripts of ≥200 nucleotides in length, have been identified in several cancers and the number keeps increasing [9]. The strategies that lead to the identification of cancer-associated lncRNAs and the methodologies involved have been comprehensively described in a review by Huarte-M [10]. Currently available evidence suggests that their activities in several biological processes are involved in the regulation of tumor cell proliferation, invasion and apoptosis [10,11,12,13]. Among lncRNA mechanisms of action, many of them function by acting as competing endogenous RNA (ceRNA), binding to and sponging specific microRNAs (miRNAs), thereby affecting corresponding miRNA-regulated downstream mRNA translation [14,15]. The role for lncRNAs in the specific biology of ES has only recently come into focus. A seven-lncRNA signature for ES prognosis, but not for the disease per se has been reported [16]. In this review, we report on those lncRNAs that have been associated with ES tumorigenesis with a focus on their prognostic and therapeutic implications.

## 2. LncRNAs and Their Biogenesis and Classification

The advent of high throughput sequencing technologies (e.g., RNA-seq) made it possible to ascertain that over 98% of the human genome is transcribed into RNA molecules that are not further translated into polypeptides and ultimately named non-coding RNAs [17]. Contrary to what was originally speculated, organismal complexity derives not from the number of coding genes that a species has, but rather in the number and diversity of non-coding genes [18]. The more complex an organism is, the greater the ratio of non-coding RNAs to mRNAs present in its genome [19]. This view led pioneers in the field to suggest that being “junk” is not synonymous to being “useless” [20,21,22]. Long noncoding RNAs (lncRNAs) are a subclass of RNA, produced during transcription by RNA polymerase II (Puli), with a limited or non-protein-coding potential [7]. However, some lncRNAs have been found to harbor short open reading frames (ORFs) that enable them to be translated, though only a limited percentage result in functional and stable peptides [23]. They are usually at least 200 nucleotides long and expressed in a tissue-specific manner [24]. Most lncRNAs are localized to the nuclear compartment, making them prone to aberrant splicing and polyadenylation, as well as resistant to exosome-mediated degradation. In the nucleus, however, they perform effective regulatory functions for nuclear organization and gene regulation [25,26]. According to current lncRNA annotations, humans have 30,000 to 60,000 lncRNA genes [27,28]. Their functions span a variety of cellular processes in development, cell proliferation and cell death including, but not limited to, transcriptional and translational regulation, post-translational modifications, and chromatin remodeling [29,30]. LncRNAs may act as miRNA decoys in order to stabilize target mRNA and to promote translation or serve as scaffolds for the assembly of protein complexes such as ribonucleoproteins (RNPs) [31]. The biogenesis of lncRNAs has great similarity to that of mRNAs [32,33]. Since lncRNAs are encoded within the genome, their transcription is in tight coordination with those of other genes, including protein-coding genes. Just like protein-coding genes, the majority of lncRNAs are transcribed by RNA polymerase II (RNA PolII) that has a high processing capability to ensure the stability and ultimate functionality of the non-coding transcript [34]. The most common method for classifying lncRNAs considers their genomic location relative to their proximity to the coding genes, as depicted in Figure 1. 

Although no consensus currently exists in regard to their general classification [35,36], lncRNAs are classified as either intergenic (i.e., those localized between two protein-coding genes, which represents the majority of lncRNAs) or intronic/exonic (i.e., those localized in the introns or exons of a protein-coding gene). These classes are further subdivided into opposite (where lncRNA is encoded on the opposite strand to a protein-coding gene) and bidirectional forms (where lncRNA is located within a promoter in the opposite direction to the encoding gene). Other lncRNA classifications have been suggested in some reviews [37,38].

### 2.1. LncRNAs Mechanisms of Action

The mechanism of action of lncRNAs depends not only on their intracellular location and the molecular partners they engage [39], but also on the developmental stage of the cell, as well as tissue specific expression. When they are localized in the nucleus, lncRNAs are capable of interfacing with different chromatin remodeling enzymes, specific transcription factors, and with RNA PolII to impact gene expression globally [39]. LncRNAs can also act in cis through modulation of transcription of closely located genes [10]. On the other hand, lncRNAs in the cytoplasm have been reported to possess the capacity of sequestering specific microRNAs (miRNAs), thus functioning as competing endogenous RNAs (ceRNAs) [40,41]. The function of lncRNAs has been stratified into four main types of molecular mechanisms described elaborately in a recent review [42]. The mechanisms include lncRNAs functioning as (1) signals for transcriptional regulation and re-programming of cells, (2) guides binding to proteins and direct them to specific sites, (3) decoys acting as endogenous target mimics to bind to intermediary regulatory molecules and sequester them away from their respective target sites, and (4) scaffolds providing structural supports for molecules to form functional complexes. Although the majority of lncRNA species are nuclear, miRNA-sponging is the most highlighted functional mechanism of lncRNAs in reports on their role in the pathogenesis of different cancers. However, it is clear that this is not the only mode of lncRNAs gene regulatory functionality as revealed by current evaluations [43]. One can therefore argue here that there is a bias in the identification of lncRNA’s mechanism of action due to the currently available limited tools we have at hand. 

### 2.2. LncRNA Expression Levels and Their Implications

The observation that a substantial fraction of the genotypic variation underpinning complex phenotypic traits occurs in non-coding regions, many of which are transcribed into distinct lncRNAs, has led to the appreciation that lncRNAs may play a central role in the molecular etiology of complex diseases [44]. Understanding lncRNA function requires thorough characterization of the molecular pathways that dictate their production, structure and turnover [45]. LncRNA expression is highly cell- and tissue-specific, and individual lncRNA expression levels could determine their suitability as either a potential biological marker for specific pathological states or a potential target for therapy [46]. The majority of lncRNAs are processed like mRNAs (i.e., undergo splicing, 5′-end capping and 3′-end polyadenylation), albeit with less efficiency. Several high-throughput studies have shown that lncRNAs are expressed at relatively lower levels than mRNAs [25,26], which is a feature that possibly underlies their late discovery. Measuring the abundance of a lncRNA is very crucial to understanding its mechanism(s) of action and function. Two recent reviews have elaborately reported the significance of lncRNA abundance to function [47,48] and discussed why it is crucial to quantitate lncRNA expression within a cell or tissue in order to shed light on their physiological/pathological roles. They also discussed methods employed to examine lncRNA and protein expression at the single cell, subcellular and suborganelle levels, the average and local lncRNA concentration in cells, as well as how lncRNAs can modulate the functions of their interacting proteins even at low stoichiometric concentrations. It was opined that the subcellular localization and the relative abundance of lncRNA determines its mode of action.

A database to enable the systematic compilation and updating of information was developed (lncRNAdb: can be accessed at http://www.lncrnadb.org/) containing a comprehensive list of lncRNAs that have been shown to have, or to be associated with, biological functions in eukaryotes, as well as mRNAs that have regulatory roles [49]. Each entry entails referenced information about RNA, including sequences, structural information, genomic context, expression, subcellular localization, conservation, functional evidence and other relevant information. They also gave an example (GAS5) that the lncRNA transcript concentrations may be controlled by RNA degradation through the non-sense mediated decay (NMD) mechanism and not by modulation during transcription. Associations between differential lncRNA expression levels and sex of patients could also predict gender predisposition for specific cancers, as was shown for the lncRNAs BLACAT2, FOXP4-AS1 and UCC in gastric cancer patients [50].

### 2.3. LncRNA Screening Strategies, Their Limitations and Prospects

Several screening strategies have been employed in identifying disease-linked lncRNAs. Methodologies employed for identifying cancer-specific lncRNAs have been comprehensively reviewed by Huarte-M [10]. LncRNAs can be detected from de novo sequencing data that have not been filtered [51]. A current review highlighted different techniques used to identify and annotate novel lncRNAs [42]. Several RNA-seq studies have helped in identifying aberrantly expressed lncRNAs across multiple diseases [52,53]. In Ewing sarcoma, the associated lncRNAs have mostly been discovered through qRT-PCR assays by screening with primers for known lncRNAs while others have been identified by bioinformatic analyses on publicly available transcriptomic data from, for example, the Gene Expression Omnibus (GEO). Given the large number of unexplored lncRNAs encoded by the genome, it goes without saying that the identification of specific ones related to specific diseases would be strongly biased by the method of discovery employed. For example, if microarray-based gene expression profiles are used, only pre-defined lncRNAs with relevant probe sets can be identified, if expressed at reasonable levels. Similarly, classical short-read RNA-seq may only detect annotated lncRNAs, while long-read sequencing technologies like Nanopore or Smart-seq may identify un-annotated lncRNAs. With the current trend in the development of high-throughput sequencing technologies, a large amount of lncRNA expression data is accumulating from different cancer samples and cell lines. 

A number of research tools have been developed to explore basic biological processes and signaling pathways regulated by lncRNAs. The LncRNAs2Pathways, for example, is a novel computational method based on a global network propagation algorithm, which could potentially enable researchers to identify signaling pathways regulated by the combinatorial effects of a set of lncRNAs [54]. An alignment-free computational tool to distinguish lncRNAs from mRNAs in RNA-seq data called PLEK (predictor of lncRNAs and mRNAs based on an improved k-mer scheme) was established by Li-A and colleagues [51]. They propose PLEK is especially suitable for SMRT long-read sequencing data and large-scale transcriptome data. Fan-XN and Zhang-SW [55] developed a powerful computational predictor program, which they named LncRNA-MFDL, to identify lncRNAs by fusing multiple features of the open reading frame, k-mer, RNA secondary structure and the most-likely coding domain sequence and using deep-learning classification algorithms where they could achieve around 97% lncRNA prediction accuracy. BASiNET (BiologicA1 Sequences NETwork) is another proposed method that is based on feature extraction from complex networks and does not require prior annotation of the genome, nor alignment of the sequences in the database [56]; it only requires nucleotide sequences in a FASTA format. BASiNET was also reported to outperform all competing methodologies and was shown to implement a supervised learning algorithm.

### 2.4. LncRNAs in Cancer

Multiple studies have reported that lncRNAs are not only involved in the initiation and progression of cancers but are highly deregulated in a multitude of tumors, acting as oncogenes or tumor suppressors [28,57,58,59,60,61,62,63,64,65]. Several lncRNAs have been shown to affect various aspects of carcinogenesis mostly through modulation of function of cancer-associated miRNAs. These functional impacts of lncRNAs in carcinogenesis have initially been appraised through knockdown/overexpression studies in cell lines and animal models. The actual list of lncRNAs that have been associated with cancer keeps expanding, as is also evidence linking their potential role as diagnostic and prognostic markers. 

## 3. LncRNAs in ES and as Potential Biomarkers

LncRNA investigations in ES are still in their infancy with just a few reports describing expression of specific lncRNAs in ES with a minimal fraction characterized for their functional relevance [29]. A comprehensive literature review was conducted by Barrett et al. [66] to identify clinically relevant ncRNAs in ES. Their study reported on the oncogenic activity of lncRNAs and miRNAs and highlighted the interplay between these two classes of ncRNAs in ES. A recent study by Chen et al. employed machine learning and training models on prevailing RNA sequencing data sets from ES patients to establish a set of seven lncRNAs that could serve as a prognostic risk marker for ES. Although their data needs experimental testing for the identified lncRNAs association and role in ES, their increased expression statistically correlated with poor overall survival [16]. LncRNAs that have so far been reported in ES are tabulated below (Table 1) and will be discussed. It must be mentioned, though, that most of these studies on lncRNAs in ES have been done using in vitro cell line models that do not fully recapitulate the physiological conditions in vivo. As such, it would be imperative to validate most of these studies using in vivo models, as suggested in a comprehensive review by Miserocchi-G and colleagues [67]. To this end, patient derived xenografts (PDX) would provide the most suitable models, as a genetic animal model for ES is still not available. 

### 3.1. EWSAT1 (Ewing Sarcoma-Associated Transcript 1)

The lncRNA Ewing sarcoma-associated transcript 1 (EWSAT1) was originally identified in Ewing sarcoma [68], and has subsequently been associated with proliferation, migration and metastases as well as overall survival (OS) in several other cancers [81,82,83]. Enhanced expression of EWSAT1 has been reported to promote cell growth, invasion and EMT by sponging specific miRNAs in various cancers [84,85]. In ES, though, Marques et al. (2014) showed that inhibition of EWSAT1 expression specifically mitigated ES cell lines capability to proliferate as well as to form colonies in soft agar. Co-expression of EWS-FLI1 and EWSAT1 in primary pediatric human mesenchymal progenitor cells (hMPCs), the most likely ES cell type of origin [4], repressed gene expression with a substantial overlap of repressed targets. Though subsequent studies supported the notion that EWSAT1 promotes ES proliferation in vitro, its gene regulatory mechanisms of action are thought to be diverse due to its nuclear and cytoplasmic localizations. They concluded that EWSAT1-mediated gene repression facilitates ES oncogenesis. 

### 3.2. HULC (Highly Upregulated in Liver Cancer)

LncRNA Highly Upregulated in Liver Cancer (lncRNA-HULC), localized on chromosome 6p24.3, is one of the few lncRNAs studied in ES. Mercatelli and colleagues [69] showed that high levels of HULC correlate with ES aggressiveness and its depletion reduces ES cell growth. They provided evidence that the sponging activity of lncRNA HULC on microRNA 186 (miR-186) results in the modulation of TWIST1 oncogene expression which impacts ES cell proliferation and clonogenicity. They also reported that treatment of ES cells with the small molecule compound YK-4-279, which targets EWS-FLI1 activity [86] resulted in both reduced HULC lncRNA expression and TWIST1 protein downregulation, ultimately sensitizing ES cells to YK-4-279 treatment. LncRNA HULC has shown to be aberrantly elevated in several tumors, including human pancreatic cancer [87], osteosarcoma [88], ovarian cancer [89] and gastric cancer [90]. Of note, in chronic myeloid leukemia (CML), it was reported that lncRNA HULC dysregulates the miR-150-5p/MCL-1 axis to impact imatinib resistance [91]. Since MCL-1 is highly expressed and shown to be a therapeutic vulnerability in ES [92], it will be interesting to interrogate the contribution of HULC on MCL-1 mediated ES cell survival.

### 3.3. MALAT1 (Metastasis Associated Lung Adenocarcinoma Transcript 1)

The lncRNA metastasis-associated lung adenocarcinoma transcript 1 (MALAT1), also known as nuclear enriched abundant transcript 2 (NEAT2), has been implicated in several tumor-associated cell behaviors [93]. Its overexpression has been shown to promote tumor cell proliferation, angiogenesis, migration and metastasis through various mechanisms involving chromatin and genomic modifications, transcriptional and posttranscriptional regulation that ultimately impact protein function [94]. In ES, MALAT1 activity has been based on few investigations. An earlier report showed that EZH2 helps EWS-FLI1 to drive tumor growth and metastasis in Ewing sarcoma [95]. The lncRNA MALAT1 was subsequently shown to interact directly with EZH2 in the TC71 and TC32 ES cell lines, suggesting it could have an oncogenic role in ES biology [70]. MALAT1 was then shown to be deregulated by Spleen tyrosine kinase (SYK) mediated signaling in ES cells and found to be transcriptionally activated through SYK/c-MYC pathway. They further showed that silencing MALAT1 in ES cells robustly induced cell apoptosis and G1 cell cycle arrest with concomitant suppression of cyclin D1 and upregulation of p27kip1 and p21cip1 levels [70]. They proposed that targeting SYK-mediated signaling could potentially represent a promising therapeutic strategy for treating ES patients. A current study by He-S and colleagues highlighted the fact that EWS-FLI1 induced tenascin-C (TNC) may regulate ES tumor progression through targeting the lncRNA MALAT1, and that this may be done by integrin α5β1-mdiated YAP activation [71].

### 3.4. DLX6-AS1 (Distal-Less Homeobox 6 Antisense RNA 1)

It has been reported in several cancers that the lncRNA DLX6-AS1 functions as an oncogene or an onco-promoting element, thereby regulating the aggressiveness and proliferation of diverse cancers [96]. In ES tissue and cells, lncRNA DLX6-AS1 was found to be significantly upregulated relative to normal tissue and cells, pointing to a potential oncogenic role of its overexpression in ES pathogenesis [72]. In the study, they also report the lncRNA DLX6-AS1 to be located in the cytoplasm of ES cells, suggesting a potential post-transcriptional regulatory role. They established that lncRNA DLX6-AS1 functions in ES cells by sponging the microRNA miR-124-3p to modulate CDK4 mRNA activity.

### 3.5. PncCCND1_B (Promoter Associated Non-Coding RNA Transcribed at the Cyclin D1 Locus)

Promoter associated non-coding RNAs (pancRNAs) transcribed at the CCND1 locus were first identified in Hela cells with pncCCND1_D being the most prominently expressed species [97]. In the search for factors that regulate CCND1 expression in Ewing sarcoma cells, Palombo-R and colleagues discovered that the lncRNA pncCCND1_B is transcribed from the CCND1 promoter region [73]. In their study, they found that in the presence of EWS-FLI1, pncCCND1_B aids DHX9 complex formation with Sam68 on the CCND1 promoter modulating CCND1 expression. Sam68 belongs to the STAR (signal transduction and activation of RNA metabolism) family of RNA-binding proteins that link signaling pathways to RNA metabolism [98]. Palombo and Paronetto have recently reported that etoposide treatment of ES cells was able to enhance pncCCND1_B expression and induce Sam68 re-localization to form a network hub on the CCND1 promoter which contributes to CCND1 downregulation [99]. In the presence of growth stimulatory signals such as IGF-1, which is known to play an important role in ES pathogenesis, the complex dissociates to allow for CCND1 promoter activation. This study highlights the complex regulation of cyclin D1 in ES cells and pinpoints the Sam68-DHX9-pncCCND1_B complex as a novel player in this pathway that could serve as a potential target for therapy.

### 3.6. FOXP4-AS1 (Forkhead Box P4 Antisense RNA 1)

The lncRNA forkhead box P4 antisense RNA 1 (FOXP4-AS1) has been shown to regulate proliferation, migration and invasion, as well as apoptosis in various cancers [100,101,102]. The only study in ES reported an upregulation of FOXP4-AS1 that correlated with poor prognosis in patients with the disease [74]. Knockdown of FOXP4-AS1 repressed growth, migration and invasion of ES cells in vitro and its overexpression had the exact opposite effects. They established that FOXP4-AS1 is predominantly localized in the cytoplasm of ES cells and may regulate their malignant phenotype by modulating the expression of thymopoietin (TMPO) through sponging the microRNA miR-298. TMPO, also known as laminar-associated polypeptide 2 (LAP2), can interact with lamins and BAF to regulate the organization of the nuclear structure and the dynamics of the cell cycle [103], and its role in cancer biology has been recently reported [104]. 

### 3.7. SOX2OT (SOX2 Overlapping Transcript)

LncRNA SOX2 Overlapping Transcript (Sox2OT) is highly expressed in several cancers and has been associated with unfavorable prognosis in those cancers where it was found to promote migration and invasion through diverse mechanisms [105,106,107]. Sox2OT has been mapped to 3q26.3-q27 chromosomal locus in humans, demonstrated to harborat least eight transcript variants and is abundantly expressed in embryonic stem cells [108,109]. Sox2OT was found to be crucial for the development and maintenance of the pluripotency of cancer stem cells (CSCs) [110], while its dysregulation was reported in several cancers including osteosarcoma, glioblastoma, lung and breast cancer, and others, where it plays an oncogenic or tumor-suppressor role [111]. Ma-L and associates found that in ES clinical samples, just like in most other cancers, Sox2OT lncRNA is highly expressed and contributes to its malignant behavior by sponging miR-363, thereby causing overexpression of FOXP4 protein levels to promote several downstream malignancy-inducing pathways [75]. 

### 3.8. HOTAIR (Hox Transcript Antisense Intergenic RNA)

In a recent elaborate review on the role of lncRNAs in rare tumors, Liguori-G and colleagues reported that among numerous cancer-associated lncRNAs, HOTAIR plays a crucial role in contributing to tumor development, metastatic progression and drug resistance [112]. In breast cancer, HOTAIR has been proven to have a prognostic value and suggested to be a potential therapeutic target [113]. In ES, HOTAIR was reported to be overexpressed and to promote malignant transformation through interaction with the histone-modifying proteins EZH2 and KDM1A (LSD1) by Siddiqui-H and colleagues [76]. They suggested HOTAIR may promote survival in EWS-FLI1 mediated transformation and also represent a potential therapeutic target given its high expression in tumors and low expression in most normal tissues.

### 3.9. TUG1 (Taurine Upregulated Gene 1)

The lncRNA TUG1 was initially identified in a genomic screen for genes in response to the taurine treatment of developing mouse retinal cells [114] and has since been found to play important regulatory functions in several cancer-associated biological processes [115]. Li-H et al. found that TUG1 was overexpressed in ES tissues and cell lines and suggested it plays a vital role in the progression of ES since inhibition of its expression reduced ES cell proliferation, migration and invasion [77]. Mechanistically, their study showed that TUG1 sponges miR-199a-3p, whose expression level they found repressed in ES cells, to enhance MSI2 expression. MSI2 belongs to the Musashi gene family and is a well-established tumor driver in the tumorigenesis of some human cancers [116]. MSI2 upregulation was also demonstrated to contribute to proliferation, migration and invasion of ES tissues and cells [77]. 

### 3.10. AK057037 (aka FEZF1-AS1)

The FEZ family Zinc Finger 1 Antisense RNA 1 (FEZF1-AS1) is a recently discovered lncRNA that has been shown to be highly expressed in several human malignancies and associated with poor prognosis [117]. It is located on chromosome 7q31.32 and was reported to play a crucial role in the proliferation, migration, invasion and Warburg effect of various tumors. In ES, FEZF1-AS1 was identified in 2013 as AK057037 by the group of Triche-T and colleagues [78] and reported to behave as an oncogene. They established its association with the PRC2 complex in ES that allows for chromatin remodeling which in turn promotes metastasis by perturbing transcription of genes involved in migration in Ewing sarcomagenesis.

### 3.11. DPP10-AS3 (Dipeptidyl Peptidase 10 Antisense RNA 10) 

Immune-associated lncRNAs have been shown to serve as prognostic biomarkers in some cancers including breast cancer, glioblastoma multiforme and bladder cancer [118,119,120]. Upon screening for prognosis-related lncRNAs in ES, Ren-EH et al. employed a machine learning-iterative lasso regression model to construct an 11-lncRNA signature [79]. This approach not only considers the prognostic information of each individual lncRNA, but also rejects redundant prognostic information, thereby maximizing the prognostic value of the lncRNA signature. DPP10-AS3 was one of the 11 identified differentially expressed immune-associated lncRNAs in ES. In that study, they also employed bioinformatics methods to explore relationships between the lncRNA signature and prognosis-associated immune cells, investigated the potential regulatory mechanisms involved, thereby providing novel research cues in the study of immune-related lncRNAs in ES. Although the relationships between the 11 lncRNAs and ES are currently unclear, their data suggest the 11-lncRNA signature has, so far, the highest relative prognostic value not affected by other clinical characteristics. Their study therefore reports the first immune-associated lncRNA signature related to ES prognosis.

### 3.12. Hdm365 (Human Double Minute Clone 365)

The human homologue of the murine double-minute 2 (mdm2) gene, HDM2, localized on chromosome 12q13-14, has been reported to be overexpressed in soft tissue sarcomas due to amplification [121]. About two decades ago, our group described a novel nuclear RNA, named hdm365, to be the major processing product of HDM2 transcripts, whose induction was observed after ectopic expression of p53 and after DNA-damaging treatment of tumor cell lines, as well as primary fibroblasts and lymphocytes [80]. Its high stress-inducible expression levels coupled with nuclear localization and absence of a corresponding protein suggested a novel RNA-based function for hdm365. Its size of 365 bases, being in the range of other lncRNAs, comprises the first five hdm2 exons, lacks polyadenylation, and after p53 induction, is detectable at the site of hdm2 transcription and processing only. We postulated that the presence of a putative 3′ terminal stem-loop structure is reminiscent of non-polyadenylated histone RNAs and U2 snRNA involved in splicing and speculated that hdm365 may contribute to p53 gene activity under stress conditions in ES. 

## 4. LncRNAs as Potential Therapeutic Targets in ES

Since several lncRNAs exhibit tissue- and cell-type specificity in both tumors and in normal tissues, they are projected as excellent druggable candidates, as well as suitable markers for diagnosis [28]. Over the past years, several studies have established a close relationship between altered lncRNA levels and cancer cell proliferation and survival. Several reasons why lncRNAs present the most interesting therapeutic targets are elaborately outlined in a review by Slaby et al. [122]. Considering the diversity in their prospective modes of action, lncRNAs can be targeted through multiple approaches that have been comprehensively reviewed by Arun and colleagues [123] and are briefly discussed below.

### 4.1. Antisense Oligonucleotides (ASOs)

ASOs are single stranded DNAs that bind to RNA via Watson-Crick base pairing. Upon binding to their target RNA, ASOs can modulate gene expression via steric hindrance, splicing alterations, initiation of target degradation via RNase H or other events. ASOs targeting different RNAs have recently entered clinical trials for various diseases including cancer [124] and are emerging as a potential therapeutic tool for targeting lncRNAs [123]. Although ASOs have high efficacy in cells, there are limitations to using them in the clinic, mainly due to in vivo toxicity and the absence of proper delivery systems that ultimately hampers tissue targeting by an adequate dose of therapeutic ASOs. Chemical modifications to improve their resistance to degradation and toxicity have contributed immensely to their success in the clinic [29]. GapmeR ASOs are RNA-DNA-RNA single-stranded oligonucleotide chains in which ribonucleotides may contain 2′-O-methoxyethyl modified sugar backbone [125], or additional modifications such as locked nucleic acids (LNAs) and S-constrained ethyl residues [126]. LNAs are single-stranded DNA fragments flanked by LNA nucleotides and bind complementarily to lncRNA providing recognition and cleavage of its target by RNase H. LNAs are reported to enhance affinity toward target RNA transcripts and acquire resistance to nucleases thereby enabling these oligonucleotides to remain functionally active for long periods in vivo [127,128].

### 4.2. RNA Interference (RNAi)

Short double-stranded RNAs are normally recognized by the RNA-induced silencing complex (RISC), resulting in base-pairing with a lncRNA/mRNA of interest that ultimately leads to argonaute degradation of the target transcript [129]. Several studies have demonstrated the successful employment of small-interfering RNAs (siRNAs) for different pathological conditions including cancer [130]. One study reported that nuclear lncRNAs were knocked down at higher levels using antisense strands while cytoplasmic lncRNAs were better knocked down using RNAi [131]. 

### 4.3. Small Molecules 

Small molecules have much more favorable pharmacokinetic properties than oligonucleotides [132]. However, targeting lncRNAs (and RNAs in general) with small molecules is still an emerging field. Compared to proteins, RNAs are less structurally diverse, highly negatively charged and possess few pockets or clefts for conventional drug binding [133]. It is challenging to design drug libraries that would potentiate RNA binding, as it is still unknown what would be the main requirements for a successful candidate. LncRNAs, like other non-coding RNAs, are able to form stable secondary and tertiary structures [134]. Small molecule inhibitors targeting these unique structural elements in lncRNAs could potentially destabilize the transcript or allosterically interfere with their protein binding to confer a therapeutic effect [123]. To date, several small molecules which selectively bind RNA motifs have been identified, mostly through high-throughput screening, despite the fact that existing drug libraries are optimized for protein binding [135]. An interesting example from the point of view of ES is targeting a triple helix encoded by lncRNA MALAT1 [136], most potently with a compound that acts through entropically driven binding deeply within the triplex. Either the RNA structure itself was disrupted to reduce MALAT1 levels, or compound binding prevented interaction of the triplex with other cofactors. There is a clear trend towards development of technologies for small molecule discovery against lncRNA targets sequestering proteins through protein-RNA interactions [132] and focusing on approaches that would deliver molecules with drug-like properties. Several groups concentrated on inhibition of the HOTAIR:PRC2 complex. In a 2015 study, promiscuous intercalator camptothecin was identified as inhibitor of HOTAIR-EZH2 complex formation, but the full potential of this strategy is yet to be proven [137]. Finally, lncRNA-HULC was demonstrated to be downregulated by YK-4-279 in ES, however, most likely as an indirect consequence of targeting EWS-FLI1 activity [69].

### 4.4. CRISPR-Cas System

Currently, the clustered regularly interspaced short palindromic repeats (CRISPR)/CRISPR associated (Cas) system, namely CRISPR/Cas, has been proven to be a very efficient gene editing tool and has predominantly been used for modifications of protein coding genes [138]. Few reports have indicated that transcriptional silencing of lncRNAs using CRISPR based approaches is feasible and will most likely be exploited in the near future for therapeutic targeting of these molecules at the transcriptional level [139,140]. Also, the recently developed RNA-targeting CRISPR-Cas13 system represents a promising approach to deplete lncRNAs with a potential for therapeutic purposes [141].

Based on their reported functions in the progression and growth of several cancers, targeting lncRNAs would be one potential approach to mitigate Ewing sarcomagenesis therapeutically.

## 5. Concluding Remarks and Future Perspectives

ES remains a very aggressive bone/soft tissue tumor in children and young adults despite advancements in chemotherapy, radiotherapy and surgical treatments. Exploiting both new therapeutic targets and prognostic biomarkers could aid in unravelling this disease at the molecular level. Mechanistic studies to clarify the molecular underpinnings of various cancers are one sure way to facilitate the development of novel targeted therapies. Though previously considered “junk sequences” in human genomes, lncRNAs have recently been proven to play crucial cellular network regulatory roles in different cancer types including ES. Therefore, studies aiming to comprehensively identify dysregulated lncRNAs in different cancers will identify novel avenues to developing lncRNA therapeutics. LncRNAs are also most likely to enable researchers to establish molecular signatures that will help in fine-tuning prognostication of disease course. Although the literature keeps growing on the biological functions of lncRNAs and their potential roles in tumorigenesis, functional studies on lncRNAs in ES are still in their infant stages and extensive investigations on a larger cohort of patient samples are needed for a successful translation into the clinical setting. In our current review, although we show that the dysregulation of a handful lncRNAs are reported in ES that could potentially serve as biomarkers for diagnosis and prognosis, validations are nonetheless needed. A larger patient cohort size is necessary to interrogate the functional implications of the lncRNA-dysregulation associated with some of the reported clinicopathological features, as well as use of the indicated lncRNAs as biomarkers and therapeutic targets in ES. These ES-associated lncRNAs have been shown to play pivotal roles in the regulation of several crucial cellular processes, including the EMT program, cell-cycle, cell viability, apoptosis and induction of stem cell properties. As such, these transcripts possess the ability to influence ES development at various points in tumor initiation to progression and metastasis establishment. It is known that lncRNAs often form secondary structures that are relatively stable, thereby facilitating their detection as free RNAs in body fluids such as blood. In the quest for developing non-invasive diagnostic tools for cancer detection, Moonmuang-S and colleagues performed a systematic review to identify 13 circulating lncRNAs (c-lncRNAs) whose altered expression in blood associate with overall survival (OS) in osteogenic tumors [142]. The potential of c-lncRNAs to be markers in ES is yet to be accomplished. It has also been suggested in a current review that lncRNAs can interact with reactive oxygen species (ROS) in cancer development and chemotherapy/radiotherapy resistance [143]. Considering the role oxidative stress plays in cancers generally, interactions between specific lncRNAs and ROS will be an intriguing axis of future exploration to illuminate aberrant biological processes. Though not fully experimentally proven, it appears that ES cells exploit fluctuations in EWS-FLI1 levels to sustain cellular plasticity as a potential mechanism for tumor progression and therapy resistance. A comprehensive understanding of the interactions between lncRNAs and EWS-FLI1 could help to design novel strategies for precise ES treatment in the future. So far, only a few lncRNAs have been described in ES and our knowledge of their biological functions is still limited. Known drugs that target RNA could provide the basis for establishing a precision lncRNA-based ES therapy, though, as already mentioned, more studies are warranted for a clinical translation here. Also, considering the inter- and intra-tumor heterogeneity of ES, we propose that lncRNA expression profiling by single-cell RNA-seq would be appropriate in not only investigating the molecular origins of the heterogeneity, but also shed some light on treatment responses. Altogether, we believe that lncRNAs will in the future serve as important tools in the early diagnosis, prognosis, as well as therapeutic targets for the clinical management of ES.

## Figures and Tables

**Figure 1 cells-11-01267-f001:**
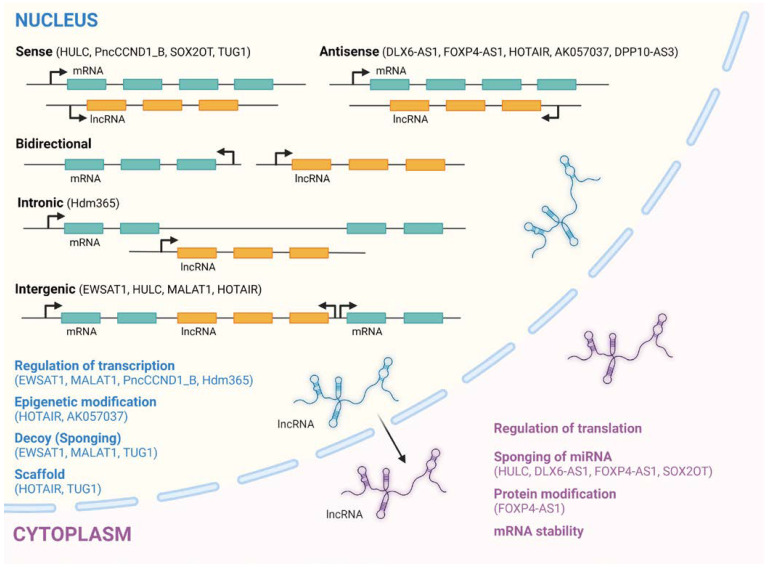
Classifications of lncRNAs and the genome locations where they are transcribed relative to protein-coding genes within the cell. Cell compartmental distribution of lncRNAs and their respective functions as well as ES-associated ones (in brackets) are indicated. Figure created with Biorender.com (accessed on 26 February 2022).

**Table 1 cells-11-01267-t001:** LncRNAs so far identified in Ewing sarcoma.

LncRNA	Expression	Method of Identification	Mechanism of Action	Targets	References
EWSAT1	Up	RNA-seq	Direct target interaction	HNRNPK	[68]
HULC	Up	qRT-PCR	Sponging miR-186-5p	TWIST1	[69]
MALAT1	Up	RNA-seq	Diverse, including Direct target interaction	EZH2, Cyclin D1, Tenascin	[70,71]
DLX6-AS1	Up	qRT-PCR	Sponging miR-124-3p	CDK4	[72]
PncCCND1_B	Up	Microarray data & RNA-seq	DHX9 & Sam68 complex formation	Cyclin D1	[73]
FOXP4-AS1	Up	Microarray data analysis	Sponging miR-298	Thymopoietin (TMPO)	[74]
SOX2OT	Up	RT-PCR	Sponging miR-363	FOXP4	[75]
HOTAIR	Up	RNA-seq	Direct target Interaction	EZH2 & LSD1	[76]
TUG1	Up	RT-qPCR	Sponging miR-199a-3p	MSI2	[77]
AK057037	Up	RNA-seq	Interaction with EZH2	PRC2 complex	[78]
DPP10-AS3	?	RNA-seq	Unclear	CD40, CD70 & CD276 molecules	[79]
Hdm365	Up upon p53 activation	Northern blot Hybridization	Hdm2 transcription & processing	P53	[80]

? = Unknown.

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
