# Peer review of "Zooming in on Long Non-Coding RNAs in Ewing Sarcoma Pathogenesis"

_cells, 2022, doi:10.3390/cells11081267_

Round 1

Reviewer 1 Report

The authors Aryee DNT et al. deeply describe the role of lncRNA in Ewing sarcoma from a biological and clinical point of view.

The manuscript is well written and complete but contains some flaws and for this reason the article needs minor revisions.

Minor concerns:

  • The authors introduce very quickly the clinical outcomes and the therapeutic management of ES and do not mention the characteristics of ES respect the other types of sarcoma. The authors should add a short description of this topic.
  • The authors describe the preclinical and clinical knowledge for each LncRNA related to ES. These paragraphs are well described but contain results obtain with preclinical models such as cell lines and primary cultures. For these reasons, I suggest to the authors to add a short description of the in vitro models used in the study described in order to clarify the concepts of the following paragraphs. For example, the following work in which the authors described the potentialities of primary cultures in cancer research should be added to the manuscript: Miserocchi G et al. “Management and potentialities of primary cancer cultures in preclinical and translational studies” J Transl Med doi: 10.1186/s12967-017-1328-z.

Author Response

We thank the reviewer for the very positive remarks on our Review manuscript and for suggesting just minor revisions. We have addressed the reviewer’s suggested points as follows:

1-“The authors introduce very quickly the clinical outcomes and the therapeutic management of ES and do not mention the characteristics of ES with respect to the other types of sarcoma. The authors should add a short description of this topic”. --- We have now added a short description as suggested as follows;

Line 34-42 Paragraph 1. ES is one of the three most common primary bone cancers in addition to osteosarcoma and chondrosarcoma. While chondrosarcoma is supposed to originate from growthplate chondrocytes and osteosarcoma from pre-osteoblasts or osteoblats, the histogenetic and developmental origins of ES are still unknown. In contrast to chondrosarcoma, which affects mostly middle-aged and older adults, and similar to osteosarcoma the peak incidence for ES is in adolescence  . In spite of the fact that ES presents a quiet genome with significant epigenetic heterogeneity different from osteosarcoma and chondrosarcoma, ES is unique on the transcriptomic level with some similarity but still distinct from mesenchymal stem cells .

2-The study by Miserocchi-G et al., that the reviewer suggested to be added to the manuscript has been added. See Line  240-246 Paragraph 3.

It must be mentioned though that most of these studies on lncRNAs in ES have been done using in vitro cell line models that do not fully recapitulate the physiological conditions in vivo. As such, it would be imperative to validate most of these studies using in vivo models as suggested in a comprehensive review by Miserocchi-G and colleagues (Miserocchi-G et al., 2017). To this end, patient derived xenografts (PDX) would provide the most suitable models, as a genetic animal model for ES is still not available.

Reviewer 2 Report

Overall, the manuscript is well written and the figure and table appear well executed. The authors should consider the following minor points below to further improve the manuscript before publication.

  1. Please prepare References according to Cells style. References must be numbered in order of appearance in the text (including table captions and figure legends) and listed individually at the end of the manuscript.
  2. Line 255. EWSAT1 and not EWSSAT1, please correct.
  3. Line 320 Paragraph 3.5 . Please describe also the regulation of pncCCND1_B expression upon genotoxic stress, in particular etoposide treatment (Palombo and Paronetto, 2022). Other promoter associated noncoding RNAs transcribed from the same i locus have been described in HeLa cells. It would be worthwhile to mention also these transcripts.

Author Response

We appreciate the reviewer’s very positive comments on our manuscript and have addressed his suggestions as follows:

1-References have been done according to Cells style.

2-Line 249; EWSSAT1 has been corrected to EWSAT1.

3-Line 308 Paragraph 3.5. Regulation of pncCCND1_B expression has been described as follows in Line  317-320 Paragraph 3.5:

Palombo and Paronetto have recently reported that etoposide treatment of ES cells was able to enhance pncCCND1_B expression and induce Sam68 re-localization to form a network hub on the CCND1 promoter which contributes to CCND1 downregulation (Palombo-R and Paronetto-MP, 2022).

Line  309-311 Paragraph 3.5 has been modified to commence with the following sentence:

Promoter associated non-coding RNAs (pancRNAs) transcribed at the CCND1 locus were first identified in Hela cells with pncCCDN1_D being the most prominently expressed species (Wang-X et al., 2008).